# The structure of the ubiquitin-like modifier FAT10 reveals an alternative targeting mechanism for proteasomal degradation

Annette Aichem[1,2], Samira Anders[3], Nicola Catone[2], Philip Rößler[3], Sophie Stotz[3], Andrej Berg [4], Ricarda Schwab[1,2], Sophia Scheuermann[1,2], Johanna Bialas[1,2], Mira C. Schütz-Stoffregen[3,5], Gunter Schmidtke[1,2], Christine Peter [4], Marcus Groettrup [1,2] & Silke Wiesner [3,5]

FAT10 is a ubiquitin-like modifier that directly targets proteins for proteasomal degradation. Here, we report the high-resolution structures of the two individual ubiquitin-like domains (UBD) of FAT10 that are joined by a flexible linker. While the UBDs of FAT10 show the typical ubiquitin-fold, their surfaces are entirely different from each other and from ubiquitin explaining their unique binding specificities. Deletion of the linker abrogates FAT10-conjugation while its mutation blocks auto-FAT10ylation of the FAT10-conjugating enzyme USE1 but not bulk conjugate formation. FAT10- but not ubiquitin-mediated degradation is independent of the segregase VCP/p97 in the presence but not the absence of FAT10's unstructured N-terminal heptapeptide. Stabilization of the FAT10 UBDs strongly decelerates degradation suggesting that the intrinsic instability of FAT10 together with its disordered N-terminus enables the rapid, joint degradation of FAT10 and its substrates without the need for FAT10 de-conjugation and partial substrate unfolding.

[1] Division of Immunology, Department of Biology, University of Konstanz, Konstanz D-78457, Germany. [2] Biotechnology Institute Thurgau at the University of Konstanz, Kreuzlingen CH-8280, Switzerland. [3] Max Planck Institute for Developmental Biology, Tübingen D-72076, Germany. [4] Computational and Theoretical Chemistry, Department of Chemistry, University of Konstanz, Konstanz D-78457, Germany. [5] Institute of Biophysics and Physical Biochemistry, University of Regensburg, Regensburg D-93040, Germany. These authors contributed equally: Marcus Groettrup, Silke Wiesner. Correspondence and requests for materials should be addressed to M.G. (email: Marcus.Groettrup@uni-konstanz.de) or to S.W. (email: silke.wiesner@ur.de)

The attachment of ubiquitin (Ub) or Ub-like (UBL) modifiers is a common post-translational modification that regulates virtually all cellular functions in eukaryotes. The UBL modifier FAT10 (also known as human leukocyte antigen-F adjacent transcript 10 or ubiquitin D (UBD)) is found exclusively in mammals[1] where it is expressed in tissues of the immune system[2–4] and upon inflammation in other cell types[2,5,6]. Importantly, FAT10 is commonly overexpressed in numerous types of cancer[3,7]. At the very C-terminus FAT10 carries a di-Gly motif that is a hallmark for the covalent attachment of Ub family modifiers to substrates through a conserved enzyme cascade. FAT10ylation is catalyzed by an activating E1 enzyme, UBA6, and a conjugating E2 enzyme, UBA6-specific enzyme 1 (USE1), that both carry a conserved catalytic residue to form a thioester intermediate with the FAT10 C-terminus[8–11]. Whether FAT10 ligation requires in vivo an E3 enzyme, as Ub and other UBL modifiers do, remains to be determined.

FAT10 is the only UBL modifier to directly target proteins for proteasomal degradation independently of Ub attachment[12–16]. Moreover, FAT10 seems to be degraded along with its substrates, since endogenous FAT10 conjugates are as short-lived as the unconjugated FAT10 monomer[14,17,18], and evidence for de-conjugating enzymes could not be obtained so far in spite of major efforts[14]. The two putative UBL domains (UBDs) of FAT10, hereafter referred to as N- and C-domain, play distinct roles in proteasome binding. The C-domain interacts with the VWA domain of the proteasome Ub receptor protein RPN10 (also called S5a in yeast). The N-domain associates with the three C-terminal Ub associated (UBA) domains of the adaptor protein NEDD8 ultimate buster-1L (NUB1L)[19,20]. NUB1L in turn also interacts with the VWA domain of RPN10 to form a ternary complex with FAT10[16].

On the sequence level, the FAT10 N- and C-domains share 29% and 36% identity with Ub for the human proteins, respectively, and only 18% identity with each other. Both mouse and human FAT10 are poorly soluble in mammalian cells and when overexpressed in *E. coli*[21,22]. This tendency to precipitate has hindered attempts to obtain a high-resolution structure of FAT10. Here, we have succeeded in increasing the solubility of full-length human FAT10 and the separate N- and C-domains. This allowed us to determine the high-resolution structures of the individual FAT10 domains by x-ray crystallography and solution-state NMR spectroscopy. Based on this structural information, we performed experiments on the role of the domain linker, the individual UBDs and the disordered N-terminal extension for FAT10 conjugation and for FAT10-mediated proteasomal degradation. Interestingly, we find that, in contrast to Ub-mediated degradation, FAT10 degradation is independent of the activity of the segregase valosin-containing protein (VCP, also known as p97 or Cdc48 in yeast)[23]. VCP enables the proteasomal degradation of poly-ubiquitylated proteins that lack loosely folded regions which can be grasped by the 19S regulator[24]. Our data suggest that the unstructured N-terminal heptapeptide together with the poor stability of FAT10 enables a fast, direct, and irreversible targeting of FAT10 along with its substrates to the 26S proteasome without the need for partial substrate unfolding by VCP and cleavage of FAT10 prior to degradation.

## Results

### FAT10 consists of two flexibly linked UBDs.

To improve the long-term stability of the wild-type (WT) human FAT10 protein, we replaced the four cysteines in FAT10 with Ser, Thr, or Leu depending on whether they were predicted by bioinformatics analysis to be solvent-exposed or buried (C7T and C9T in the N-domain; C134L and C162S in the C-domain) (Fig. 1a). To gain structural information on FAT10, we expressed and purified $^{15}$N-labeled human Cys-free FAT10 (FAT10 C0; amino acids 1–165). The protein was highly soluble and yielded a well-resolved $^{1}$H,$^{15}$N-correlation (HSQC) NMR spectrum (Fig. 1b) demonstrating the structural integrity of the Cys-free FAT10 mutant. For this FAT10 construct, we were able to obtain almost complete amide resonance assignments with the exception of the N-terminal residues (amino acids 1–6), two residues in the N-domain (amino acids 53–54), and part of the linker region (amino acids 85–87). Moreover, we detected only few amino acids in the FAT10 C0 protein that gave rise to small magnitude (<0.2) steady-state {$^{1}$H}-$^{15}$N NOEs (Supplementary Fig. 1a). These included the C-terminal residues (amino acids 162–165), S84 in the linker region and five peaks for which we were unable to obtain resonance assignments due to line broadening. This demonstrates that FAT10 contains an N- and a C-terminal tail that are both disordered and two structured domains that are joined by a flexible linker.

For further structural studies, we designed a Cys-free FAT10 construct lacking the N-terminal four residues (ΔN FAT10; amino acids 5–165) and Cys-free constructs of the individual N- and C-terminal domains (amino acids 5–86 and amino acids 85–165, respectively; hereafter referred to as N- and C-domain) (Fig. 1a). To assess whether the N- and C-domains interact with each other in FAT10, we compared the $^{1}$H,$^{15}$N-HSQC spectra of ΔN FAT10 with those of the individual domains (Fig. 1c). This type of NMR spectra gives rise to one peak per H$^{N}$–N atom pair translating essentially into one peak per amino acid in proteins. Since the position of NMR signals (chemical shift) depends directly on the local chemical environment of the observed atomic nuclei, chemical shift changes (perturbations) are highly sensitive reporters on protein interactions and conformational changes. Interestingly, an overlay of the ΔN FAT10, N- and C-domain $^{1}$H,$^{15}$N-HSQC spectra exhibited only very limited changes in peak positions (Fig. 1c). The most notable chemical shift perturbation was observed for E86 that resides at the very C-terminus of the N-domain and is the second N-terminal residue in the C-domain, while it is embedded in the linker region in the ΔN FAT10 construct. The chemical shift changes for E86 thus arise from the different chemical environments of this residue in the ΔN FAT10, N-, and C-domain constructs. Otherwise, almost all peaks of the N- and C-domain, respectively, superimpose well with the ΔN FAT10 construct (Fig. 1c).

Taken together, these NMR analyses show that FAT10 contains two independently folded domains that are joined with a flexible linker and additional flexible regions at the N- and C-terminus.

### The FAT10 domains both adopt Ub-like folds.

To characterize the structures of the FAT10 domains in more detail, we crystallized the N-domain and solved the solution NMR structure of the C-domain, since we could not obtain crystals of the latter. Both domains adopt the typical ββαββαβ β-grasp UBL fold (Fig. 1d, e). The N-domain crystallized readily and we solved its structure at 1.95 Å resolution by molecular replacement (Fig. 1d, Supplementary Table 1). We could resolve three virtually identical protein chains in the asymmetric unit that superimposed with a backbone root mean square deviation (r.m.s.d.) of 0.66 ± 0.01 Å for amino acids 8–81 (Supplementary Fig. 1b). Only the N- and C-terminal residues (amino acids 6–7 and 82–84, respectively) display distinct conformations and increased B factors showing that these residues are disordered and not part of the UBL fold. Of note, the side chains of residues 7 and 9 that are cysteines in the native human FAT10 sequence are indeed solvent exposed and only residue 9 is part of the UBL fold. Importantly, superposition of our crystal structure with the NMR structure of the human FAT10 N-domain (PDB-ID: 2MBE)[25] revealed significant differences in the orientation of the α1 helix (Supplementary

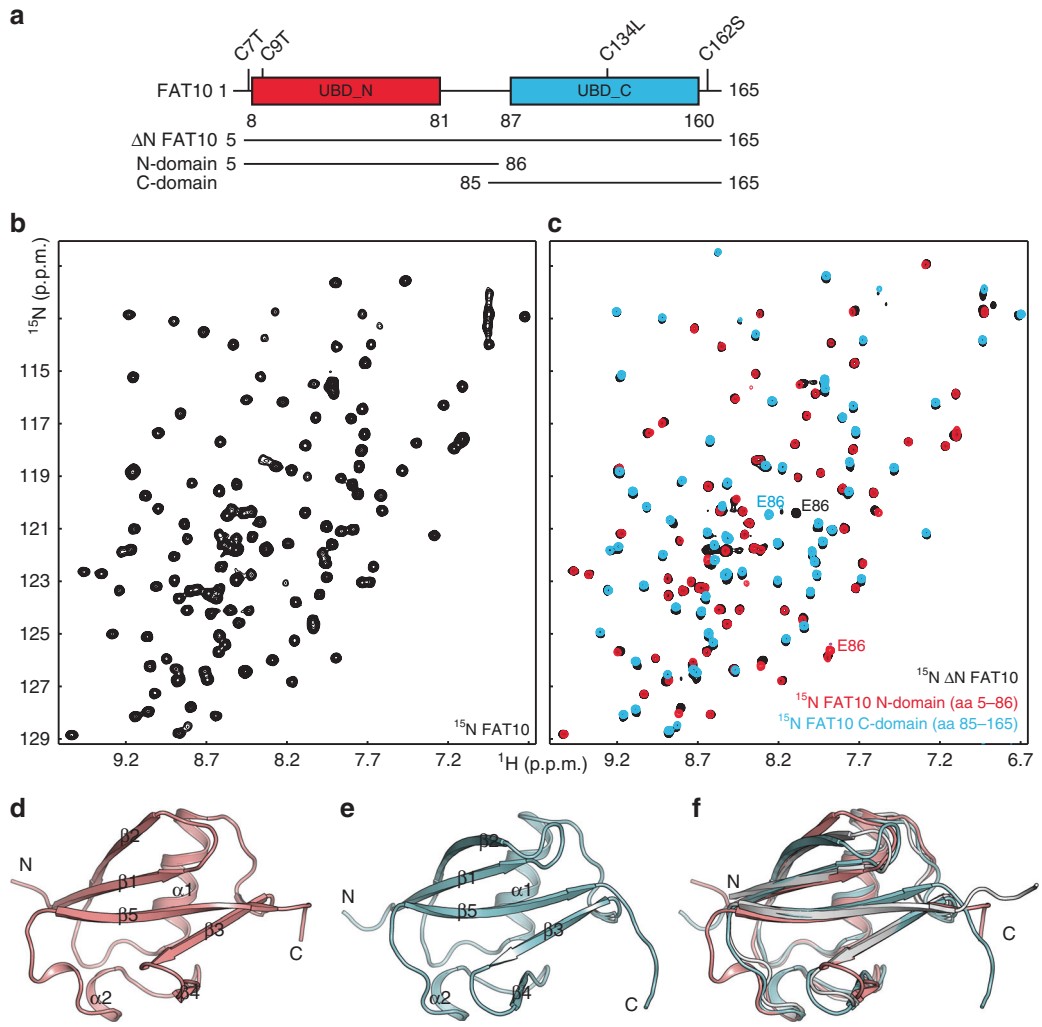

**Fig. 1** The two FAT10 UBDs are structurally independent in solution. **a** Schematic representation of the FAT10 NMR constructs and domain boundaries. Ubiquitin-like domain (UBD)_N and UBD_C denote the boundaries of the N- and C-terminal ubiquitin-like (UBL) folds, respectively. **b** Representative region of the $^1$H,$^{15}$N-correlation spectrum of Cys-free full-length human FAT10 (FAT10 C0) and **c** an overlay of the $^1$H,$^{15}$N-correlation spectra of the Cys-free FAT10 lacking the N-terminal four amino acids (ΔN FAT10) with the isolated Cys-free FAT10 N- (red) and C- (cyan) domains, respectively. Except for E86 at the C-terminus of the N-domain and the N-terminus of the C-domain, the spectra of the isolated domains are highly similar to the ΔN FAT10 construct demonstrating the structural independence of the domains. **d** Ribbon representation of the crystal structure of the FAT10 N-domain. Secondary structure elements are labeled to highlight the canonical ββαββαβ β-grasp fold. **e** Ribbon representation of the lowest energy NMR structure of the FAT10 C-domain. **f** Overlay of the structures of the FAT10 N- and C-domain (salmon and cyan, respectively) with Ub (PDB ID: 1UBQ) (gray). The structures superimpose with Ub with an r.m.s.d. for all backbone atoms of 0.764 Å for the N-domain (amino acids 8–81) and 0.964 Å for the C-domain (amino acids 88–160)

Fig. 1c), but almost perfectly aligned with the crystal structure of Ub (PDB-ID: 1UBQ)[26] (Fig. 1f). As in Ub, the core of the UBL fold is stabilized by a dense hydrophobic interaction network that is almost void of aromatic residues. Overall, comparison of the FAT10 N-domain structure with Ub showed that not only the structured regions are well-conserved, but also the lengths of the loops except for the α2–β5 loop that contains two additional residues in the N-domain (Figs. 1f and 2a).

For the C-terminal domain, crystallization trials were unsuccessful. We therefore determined the three-dimensional structure of this FAT10 domain using standard triple resonance solution NMR experiments (Fig. 1e, Supplementary Table 2). The final ensemble of structures superimpose well within the UBL fold (backbone r.m.s.d. of 0.27 ± 0.03 Å for amino acids 87–160), while the N- and C-terminal regions (amino acids

85–86 and 161–165, respectively) are disordered as in the case of the N-domain (Supplementary Fig. 1d). Of note, the side-chain of amino acid 162 that is a Cys in the native human FAT10 sequence is solvent exposed and not part of the UBL fold, while residue 134 is indeed buried in the hydrophobic core validating our substitution to a Leu. As for the N-domain, the structure of the C-domain is highly similar to Ub (Fig. 1f) and differs only in the length of the β1–β2 loop that comprises two additional residues (Figs. 1f and 2a).

**The surfaces of the two FAT10 UBDs and Ub differ.** With the structures of the FAT10 N- and C-domains in hand, we performed a structure-based sequence alignment with Ub (Fig. 2a) and compared the protein surfaces of Ub and the FAT10 N- and

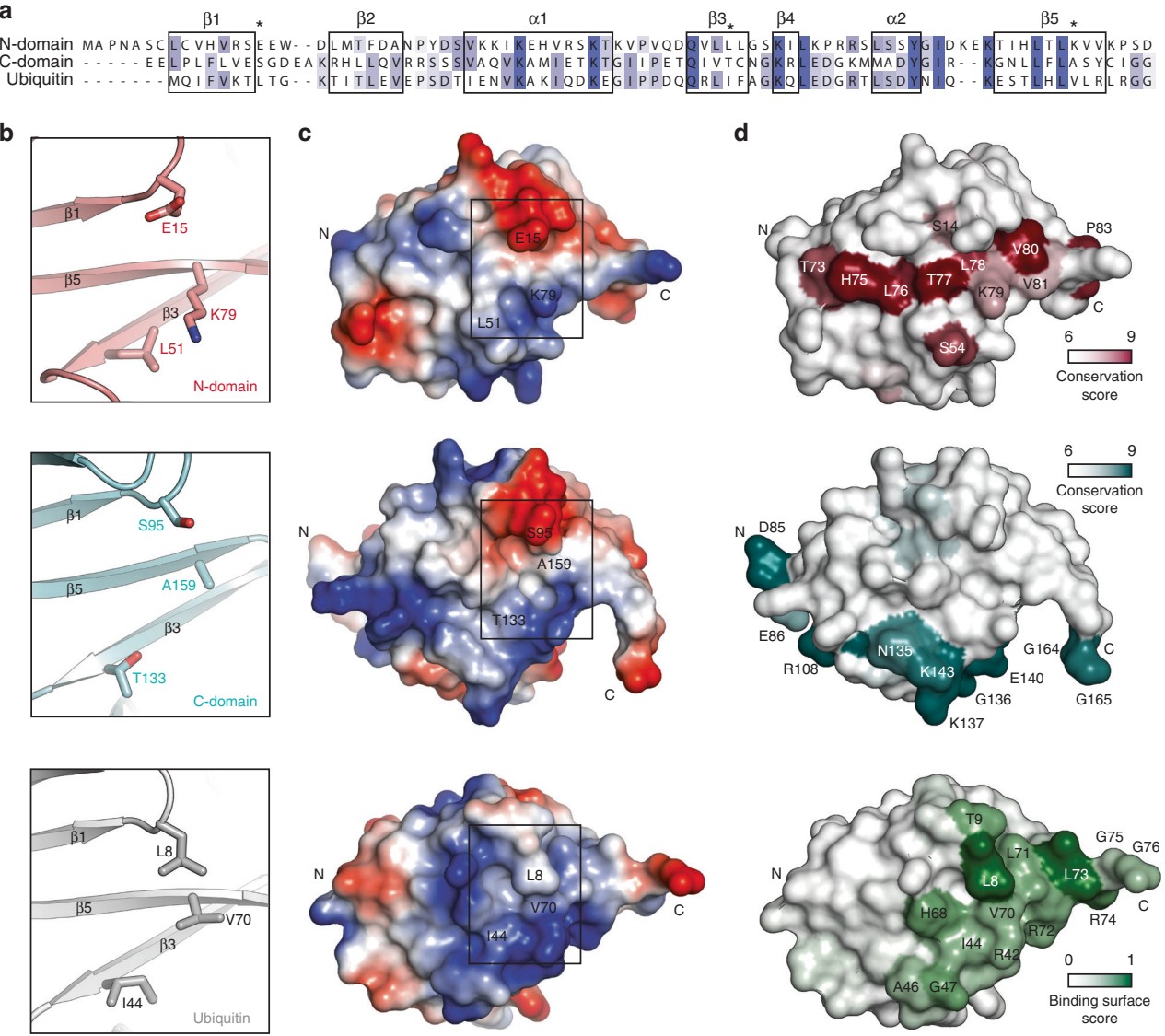

**Fig. 2** The surfaces of the FAT10 UBDs are distinct from Ub and each other. **a** Structure-based sequence alignment of the human FAT10 domains with Ub. Residues are color coded by sequence identity from white (0%) to dark blue (100%). Secondary structure elements are labeled and boxed. The residues corresponding to the hydrophobic patch in Ub are marked with asterisks. **b** Close-up on the residues equivalent to the hydrophobic patch in Ub showing the N-domain (top), the C-domain (center), and Ub (bottom). **c** Electrostatic surface potentials of the FAT10 N- (top) and C-domains (center) and Ub (bottom). Negatively charged areas are colored in red, while positively charged regions are in blue and neutral residues in white. The position of the hydrophobic patch residues in Ub and equivalent positions in the FAT10 domains are labeled. The boxes outline the close-up regions depicted in **b**. **d** Surface plots of the FAT10 N- (red; top) and C-domains (teal; center) colored by conservation of residues in mammalian FAT10 proteins (Supplementary Fig. 4). Highly conserved residues are labeled. The surface of Ub is color coded in green by occurrence of residues at the interfaces of Ub–protein complexes (bottom). All structure and surface representations are in the same orientation as in Fig. 1d–f

C-domains (Fig. 2b–d, Supplementary Fig. 2). Interestingly, the hydrophobic patch in Ub (L8, I44, and V70) that is used for most Ub–protein interactions[27], is neither conserved in the FAT10 N-domain (E15, L51, and K79) nor in the C-domain (S95, T133, and A159) (Fig. 2b, Supplementary Fig. 2a). Moreover, Ub and the FAT10 N- and the C-domains have distinct electrostatic surface potentials (Fig. 2c, Supplementary Fig. 2b). Consistent with these surface properties and in agreement with previous studies[16,28], we observed no chemical shift perturbations in the $^1$H,$^{15}$N-HSQC spectra of the ΔN FAT10 construct when we added the unlabeled second Ub interacting motif (UIM2) of the proteasomal adapter protein RPN10, while Ub readily interacted with the UIM2 as expected (Supplementary Fig. 3).

In contrast to Ub, the sequences of FAT10 proteins have evolved considerably during mammalian evolution. To compare the FAT10 N- and C-domains in more detail and to identify conserved regions, we generated a multiple sequence alignment of FAT10 proteins (Supplementary Fig. 4). This revealed a conserved surface in the FAT10 N-domain that mainly consisted of the β5 strand (Fig. 2b, d, top panels, Supplementary Fig. 2a, c, top panels). Interestingly, this region partially overlaps with the region surrounding V70 in Ub that is essential for many Ub–protein interactions (Fig. 2b, d, top and bottom panels)[29]. In contrast, conserved surfaces on the FAT10 C-domain included those next to the essential C-terminal di-Gly motif, the region around the β4 strand, and the β3–β4 and β4–α2 loops (Fig. 2d, center panel, Supplementary Fig. 2a, c, center

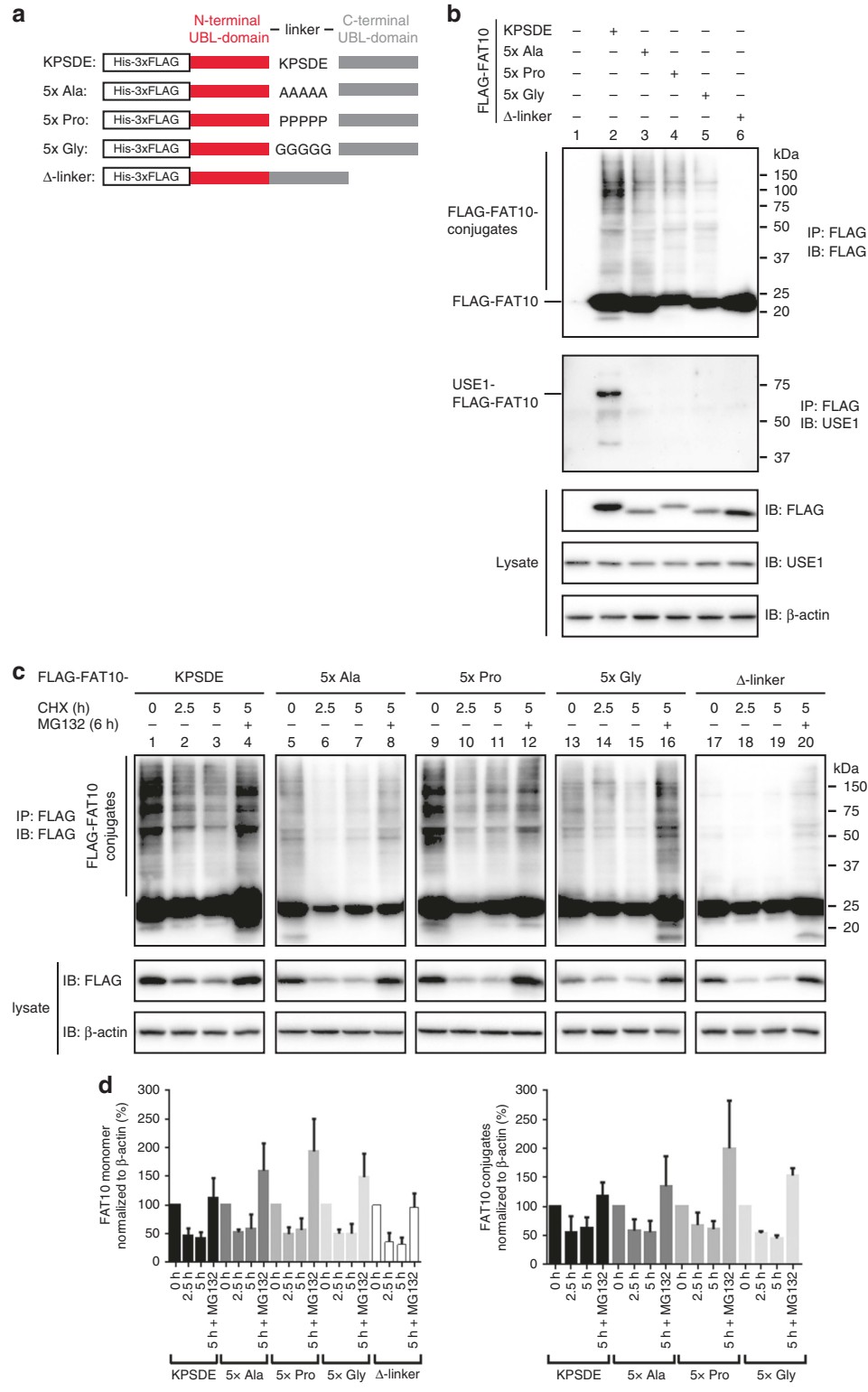

panels), that partially overlapped with the area in vicinity to I44 in Ub (Fig. 2b, d, center and bottom panels). Moreover, the conserved surface patch in the FAT10 C-domain contained hardly any positions that were conserved in the FAT10 N-domain (Fig. 2d, top and center panels, Supplementary Fig. 2c, top and center panels).

Together with previous functional studies[16], our structural and sequence analyses show that the FAT10 domains and Ub possess individual binding specificities that stem from their distinct protein surfaces. In support of this, the spindle checkpoint protein Mad2 interacts with the FAT10 N-domain, but not the C-domain, on a surface that includes the highly conserved β5 strand[25].

**The FAT10 linker is essential for activation and conjugation.** Our structural studies revealed that the two FAT10 UBDs are

**Fig. 3** The presence of the FAT10 linker is crucial for FAT10 conjugation. **a** Scheme showing the amino acid substitutions or deletion of the WT FAT10 linker sequence (KPSDE). The different linker mutants 5x Ala, 5x Pro, 5x Gly, and Δ-linker were generated by site-directed mutagenesis of the expression plasmid pcDNA3.1-His-3xFLAG-FAT10. The His-3xFLAG tag is referred to as FLAG in the text as well as in the figures. **b** HEK293 cells were transiently transfected with expression constructs for the indicated proteins to monitor proteome-wide and USE1-specific conjugation of WT as well as of mutant FAT10. Uncropped blots are shown in Supplementary Figure 8. **c** The degradation rate of monomeric FAT10 in the lysate as well as of FAT10 conjugates was monitored in HEK293 cells expressing the different FLAG-tagged FAT10 linker variants. Cells were treated for 2.5 or 5 h with 50 µg/mL CHX to inhibit de novo protein synthesis. Where indicated, cells were additionally treated with 10 µM proteasome inhibitor MG132 for 6 h. **b, c** HEK293 cells were harvested and lysed 24 h after transfection. Cleared protein lysates were subjected to immunoprecipitation using EZview Red anti-FLAG-M2 affinity gel. Proteins were separated on 12.5% Laemmli gels and visualized by western blot analysis with a monoclonal FLAG-reactive antibody (clone M2) or a USE1-reactive polyclonal antibody, as indicated. β-actin was used as loading control. One experiment out of three experiments with similar outcomes is shown. **d** Quantification of the amount of monomeric and of conjugated FLAG-FAT10 linker variants in **c**. The enhanced chemiluminescence (ECL) signals were quantified with Image Lab 4.1. software (BioRad) and normalized to signals of the loading control β-actin. The values of the untreated cells were set to 100% and the other values were calculated accordingly. Values are shown for three independent experiments with similar outcomes as means ± s.e.m.

tethered by a flexible linker that comprises amino acids 82–86 (KPSDE). To address the functional importance of this linker, we replaced the five WT amino acids (KPSDE) of the linker with five alanines (5x Ala), five prolines (5x Pro), or five glycines (5x Gly), or deleted the entire linker (Δ-linker) in a FLAG-tagged FAT10 construct (Fig. 3a and Supplementary Fig. 5). To investigate the conjugation efficiency and stability of these FAT10 variants, we transiently transfected HEK293 cells with FLAG-tagged WT FAT10 or the different FAT10 variants (Fig. 3b). USE1 is a bis-pecific E2 enzyme for Ub and FAT10 that undergoes auto-FAT10ylation in vivo (Fig. 3b, lane 2)[11]. However, the functional consequences of this modification are unclear. We found that linker mutations did not significantly affect the level of proteome-wide FAT10 conjugation, but abrogated USE1 auto-FAT10ylation (Fig. 3b, lanes 3–5). Of note, CRISPR/Cas9-mediated knockout of USE1 in HEK293 cells confirmed USE1 as the only E2 for FAT10, since FAT10 conjugation was absent in these cells (Supplementary Fig. 6). The absent USE1 auto-FAT10ylation, but unaffected bulk conjugation activity of the linker mutants (Fig. 3b) thus demonstrate that auto-FAT10ylation of USE1 is not essential for USE1 function with respect to FAT10 conjugation activity.

In contrast to the linker mutants, deletion of the linker abolished FAT10 conjugation (Fig. 3b, lane 6). We confirmed this result in an experiment that reported exclusively on isopeptide-linked conjugates, but not on non-covalent interactions (Supplementary Fig. 5b, lanes 2–6) and thus cannot form FAT10 conjugates. We therefore conclude that deletion, but not mutation, of the FAT10 linker abrogates FAT10ylation activity.

To investigate the role of the linker in FAT10 degradation, we performed cycloheximide (CHX) chase experiments with FLAG-tagged WT FAT10 or the different linker mutants by treating HEK293 cells with CHX to prevent protein de novo synthesis and, where indicated, additionally with the proteasome inhibitor MG132 (Fig. 3c). Since the different FAT10 variants were expressed at different levels (Fig. 3b, c), we normalized the signal intensities of monomeric FAT10 and of the FAT10 conjugates to the respective β-actin signals in the lysate (Fig. 3d). All monomeric FAT10 variants (Fig. 3d, left panel) and their conjugates (Fig. 3d, right panel) were similarly degraded within 2.5 h of CHX chase and their degradation could be blocked by proteasome inhibition. As before (Fig. 3b, lane 6), the Δ-linker FAT10 was incapable of forming conjugates (Fig. 3c, lanes 17–20), since the background signals for the Δ-linker mutant were absent under denaturing conditions (Supplementary Fig. 5b, c) and thus resulted from non-covalent protein interactions.

Taken together, these assays demonstrate that mutation of the linker abolishes FAT10ylation of specific substrates such as USE1, but does not affect bulk FAT10 conjugation or FAT10 stability. We also conclude that auto-FAT10ylation of the E2 enzyme USE1 is not required for its activity in vivo.

**The folds of the FAT10 UBDs are less compact than in Ub**. In contrast to Ub, monomeric FAT10 is short-lived and directly degraded by the proteasome. With a half-life of only 1 h[14], FAT10 is degraded substantially faster than almost all other cellular proteins[30]. Moreover, the WT FAT10 protein is prone to pre-cipitation when expressed in eukaryotic cells or in E. coli[21,22]. During our NMR studies, we observed that replacement of the FAT10 Cys residues could alleviate this problem. To investigate whether this behavior coincides with a loose fold of the FAT10 UBDs, we performed differential scanning fluorimetry (DSF) experiments and determined the melting temperatures of WT ΔN FAT10, the C0 ΔN FAT10 mutant, and Ub (Fig. 4a, Supplementary Fig. 7a). This revealed that the WT ΔN FAT10 and the C0 ΔN FAT10 mutant melted already at 41 °C and 47 °C, respectively, while Ub had a melting temperature of 83 °C. In support of this, we observed elevated atom fluctuations and lower average secondary structure contents throughout the entire N- and C-domain of FAT10 in MD simulations as compared to Ub (Supplementary Fig. 7b). Taken together, this shows that the FAT10 N- and C-domains are more loosely folded than in Ub and thus may be less stable in cells.

**Substitution of FAT10 UBDs by Ub impedes degradation**. To investigate whether the poor solubility of FAT10 and the less tight folding of its UBDs may facilitate the joint degradation of FAT10 and its isopeptide-linked substrates and to examine the functional differences of the individual UBDs in FAT10 conjugation and stability, we created FLAG-tagged FAT10-Ub chimeras where we replaced either the N- or C-domain with Ub (FLAG-Ub-FAT10 or FLAG-FAT10-Ub, respectively) (Fig. 4b). For both substitutions, we used Lys-free Ub to prevent Ub chain formation. Moreover, for N-domain substitution the Ub lacked the C-terminal di-Gly motif to prevent cleavage of the chimera by Ub-specific isopeptidases. To monitor FAT10ylation activities, we co-transfected HEK293 cells with HA-tagged USE1 and FLAG-tagged WT FAT10, the chimeric proteins, or FLAG-tagged WT Ub. Both FAT10-Ub hybrid proteins were conjugated to substrates at levels similar to WT FAT10, but both showed defects in their ability to covalently modify USE1 (Fig. 4c). Moreover, compared to WT FAT10, conjugates of the FLAG-Ub-FAT10 chimera but not the FLAG-FAT10-Ub chimera were degraded slower in CHX chase experiments (Fig. 4d, e), suggesting that the tightly folded Ub cannot fully adopt the function of the FAT10 N-domain for enabling effective proteasomal degradation.

**Substitution of cysteines stabilizes FAT10 and its conjugates**. For our structural studies, we succeeded in improving the long-term stability of FAT10 by replacing its four Cys residues. To investigate whether these Cys substitutions stabilize FAT10 in

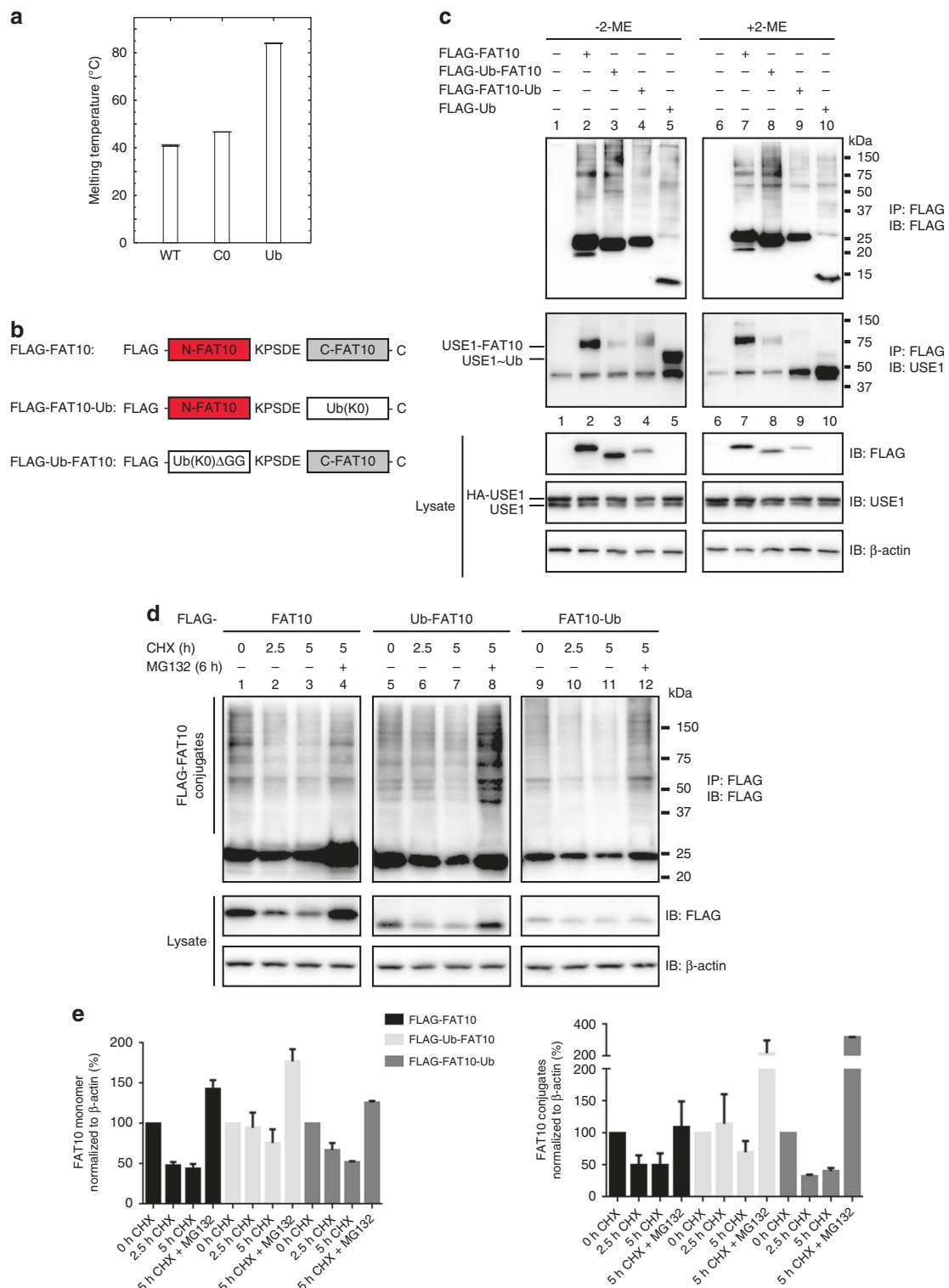

cellulo, we compared the degradation rates of FLAG-tagged WT FAT10 and the Cys-free mutant with the C134L mutations in the C-domain (FLAG-FAT10-C0-C134L) (Fig. 5a). For both WT and Cys-free FAT10, proteasome inhibition led to an accumulation of FAT10 monomer and conjugates in CHX chase experiments (Fig. 5a; lanes 6 and 11). However, Cys-free FAT10 monomer and

its conjugates were degraded much slower than WT FAT10 and its conjugates (Fig. 5a, b).

To gain insight into the role of the four FAT10 Cys residues in protein stability we performed molecular dynamics (MD) simulations of the WT and Cys-free FAT10 N- and C-domains (Fig. 5c). We observed that in general the structures of both the

**Fig. 4** The FAT10 UBDs are loosely folded and degraded less efficiently when replaced by Ub. **a** Melting temperatures are plotted in a bar diagram for WT ΔN FAT10, the C0 ΔN FAT10 mutant, and ubiquitin and were obtained from the respective maxima of the first derivative curves of three independent differential scanning fluorimetry measurements where the temperature was varied in 0.036 °C steps from 20 to 95 °C. The data represent mean ± standard deviations (s.d.). The individual data points are shown in Supplementary Fig. 7a. **b** Schematic presentation of WT FLAG-FAT10 and the two FAT10-ubiquitin hybrids (FLAG-FAT10-Ub and FLAG-Ub-FAT10), where either the N- or the C-domain of FAT10 was exchanged with lysine-less Ub (K0) to prevent Ub chain formation. To suppress cleavage of Ub(K0) from the FAT10 C-domain in Ub-FAT10, Ub(K0) was expressed without the di-glycine motif. **c** Western blots showing bulk conjugates of WT FAT10, Ub-FAT10 or FAT10-Ub proteins, and USE1 auto-FAT10ylation. **d** Degradation of monomeric WT FAT10, Ub-FAT10, and FAT10-Ub was monitored by transient transfection of HEK293 cells with the indicated expression constructs. Cells were treated for 2.5 or 5 h with 50 μg/mL CHX to inhibit de novo protein synthesis. Where indicated, cells were additionally treated with 10 μM proteasome inhibitor MG132 for 6 h. **c, d** HEK293 cells were harvested and lysed 24 h after transfection. Cleared protein lysates were used for immunoprecipitation using EZview Red anti-FLAG-M2 affinity gel. The upper panels show the bulk conjugates in the immunoprecipitation, lower panels show the expression of the proteins in the lysates. β-actin was used as loading control. One representative experiment out of three experiments with similar outcomes is shown. **e** ECL signals were quantified and graphs show the amount of monomeric as well as of conjugated FLAG-FAT10 and FLAG-tagged FAT10-Ub chimeras in the cells. The ECL signals were quantified with Image Lab 4.1. software (BioRad) and normalized to signals of the loading control β-actin. The values of the untreated cells were set to 100% and the other values were calculated accordingly. Shown are the values of three independent experiments with similar outcomes as means ± s.e.m.

WT and Cys-free domains remained folded but that the Cys-free N-domain showed a reduced flexibility compared to the WT N-domain that may lead to a net stabilization. Together with the fact that FAT10 has a melting temperature of only 41 °C (Fig. 4a, Supplemetary Fig. 7a), this supports the idea that the native FAT10 protein may be rather unstable and thus prone to fast and direct proteasomal degradation.

**The N-terminus of FAT10 renders its degradation VCP independent.** Several UBL modifiers, such as FAT10, contain N-terminal extensions of the UBL fold. However, the functions of these extensions are poorly understood. Our NMR analyses and MD simulations showed that the N-terminal tail of FAT10 is intrinsically disordered. Interestingly, recent studies suggest that unstructured extensions in substrates may play a role in bypassing the requirement for VCP/p97-mediated unfolding activity for the proteasomal degradation of tightly folded, ubiquitinated proteins[24]. To explore whether the degradation of FAT10 conjugates depends on VCP activity, we transiently transfected HeLa cells with HA-tagged Ub-(AV)-GFP (Fig. 6a) and FAT10-(AV)-GFP (Fig. 6b) as model substrates because the Deshaies group has previously shown that the unfolding of Ub-(G76V)-GFP depends on VCP activity[31]. In both proteins, the C-terminal di-Gly motif was substituted with AV to prevent cleavage by isopetidases. To monitor the degradation rates of the model substrates, we performed radioactive pulse chase experiments in the absence or presence of the highly specific and well-characterized VCP inhibitors CB-5083[32] and NMS-873[33] (Fig. 6a, b).

Remarkably, while both VCP inhibitors completely arrested the degradation of Ub-(AV)-GFP (Fig. 6a), they had virtually no effect on FAT10-(AV)-GFP degradation (Fig. 6b), although both fusion proteins were rapidly degraded in the absence of VCP inhibitors. Consistently, CHX chase experiments in HEK293 cells transiently expressing untagged FAT10 in the absence or presence of the VCP inhibitor CB-5083[32] demonstrated that the degradation of untagged monomeric FAT10 was independent of VCP activity, since in contrast to the proteasome inhibitor MG132 addition of CB-5083 did not stabilize FAT10 levels (Fig. 6c, lanes 2–6). On the contrary, degradation of a FAT10 variant that lacked the N-terminal disordered tail (FAT10-ΔAPNASC) depended on VCP activity (Fig. 6c, lanes 7–11), since addition of the VCP inhibitor CB-5083 substantially increased FAT10 protein levels for this FAT10 variant but not for WT FAT10 (Fig. 6c, d, lane 6 vs. 11). We thus conclude that, in contrast to Ub, FAT10 degradation does not require VCP unfolding activity and that the

unstructured N-terminal tail of FAT10 is a primary determinant for rendering FAT10 VCP-independent.

Of note, extended MD simulations of the N-terminal 86 amino acids of FAT10 corroborated our findings that the N-terminal tail (amino acids 1–7) of FAT10 is disordered, highly flexible, and only loosely contacts the N-domain (Fig. 6e). Taken together, our experiments demonstrate an alternative proteasome targeting mechanism where an unstructured N-terminal heptapeptide extension circumvents the requirement of unfolding activity by VCP prior to degradation by the 26S proteasome.

**Discussion**

FAT10 is the only UBL modifier that directly targets its conjugation substrates for degradation by the 26S proteasome. This finding raises the question why in addition to the Ub–proteasome system the FAT10–proteasome system has evolved in mammals. In fact, there are many striking differences between these two systems. The proteome of FAT10 conjugation substrates has the smallest overlap with that of Ub when compared to other UBL modifiers[34]. While the Ub system is constitutively active in all cell types, FAT10 is only expressed in the immune system or strictly dependent on cytokine induction in other cell types[2–4]. Ub is recycled at the 26S proteasome and hence long-lived in cells, whereas FAT10 is degraded along with its substrates and short-lived. Lastly, the Ub protein sequence is strictly conserved except for a few amino acids from yeast to humans, while FAT10 has evolved rapidly despite being present only in mammals (Supplementary Fig. 4).

The three-dimensional structures of the FAT10 domains (Figs. 1 and 2) determined in this study reveal further differences between FAT10 and ubiquitin, in spite of both of them consisting of ubiquitin fold domains. Ub and the N- and C-domains of FAT10 each have unique surface properties (Fig. 2, Supplementary Fig. 2) that enable them to interact with distinct binding partners (Supplementary Fig. 3)[16,20]. Consistent with their distinct surface properties, we found that conserved surfaces map to different regions on the FAT10 N- and C-domain structures (Fig. 2d). Together with the two FAT10 domains being structurally independent and joined by a flexible linker, this provides a structural basis for the finding that the two FAT10 UBDs can dock to different reader domains and thereby link two FAT10 binding complexes as for example RPN10 and NUB1L at the proteasome[16,20]. The importance of an intact flexible linker in FAT10 is underscored by our finding that deletion of the linker abrogates FAT10 conjugation in vivo (Fig. 3).

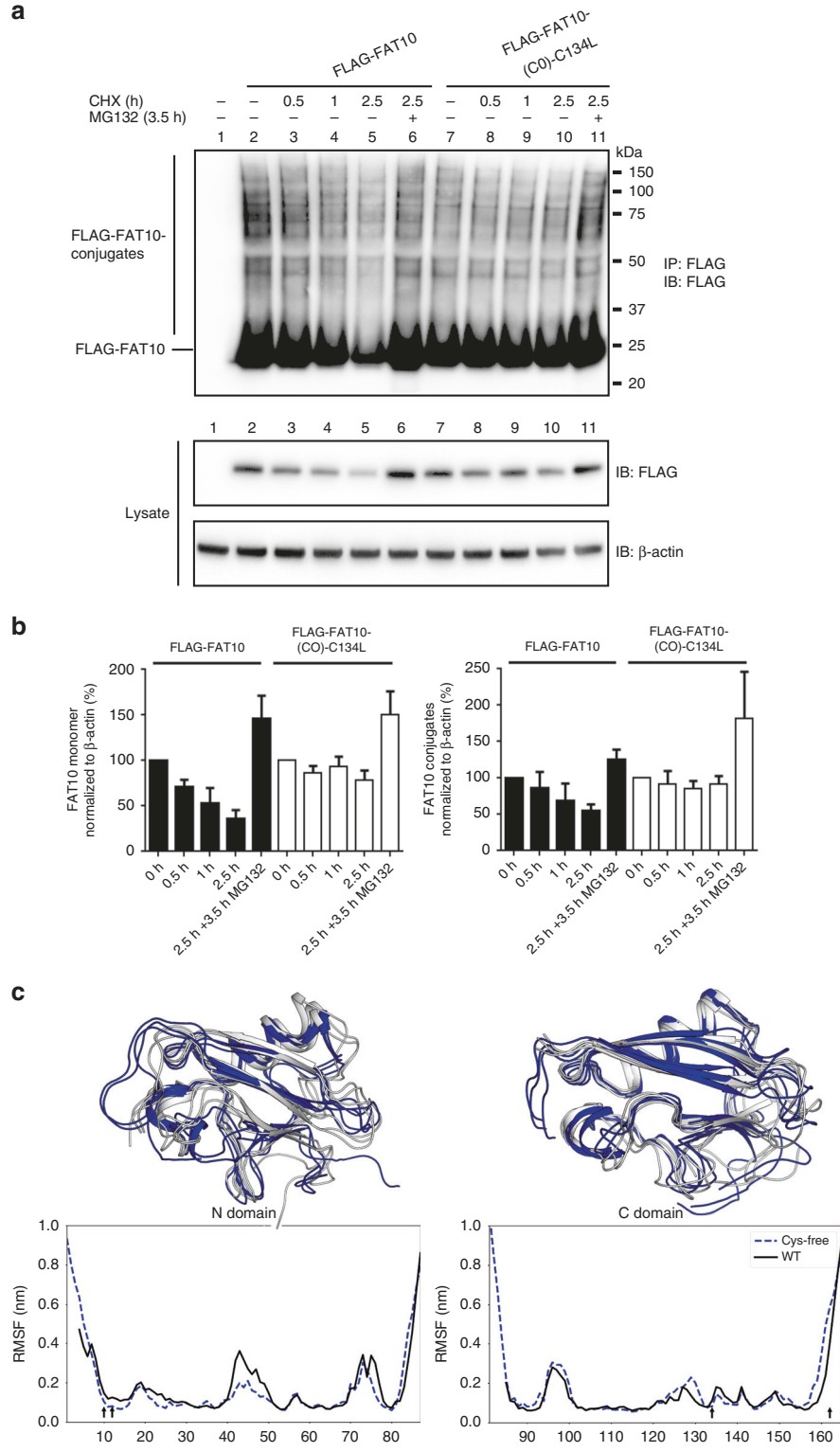

The FAT10 E2 enzyme USE1 undergoes prominent auto-FAT10ylation in vivo (Figs. 3b and 4c)[11,18], but the functional importance of this modification for FAT10 conjugation has been unclear. We found that the native protein sequence in the FAT10 linker as well as the presence of the N- and C-domain are essential for USE1 auto-FAT10ylation. However, despite the absent USE1 auto-FAT10ylation, FAT10 conjugation or the stability of FAT10 conjugates were unaffected on a proteome-wide level upon

mutation of the FAT10 linker sequence (Fig. 3). This shows that auto-FAT10ylation of USE1 per se is not required for USE1 function and FAT10 conjugation.

A major difference between Ub- and FAT10-mediated proteasome targeting is the cleavage and recycling of Ub from its substrates at the proteasome, while FAT10 seems to be degraded along with its substrates[14]. Moreover, with a half-life of only one hour in cells, the life-time of FAT10 is much shorter

**Fig. 5** Cys-replacement stabilizes FAT10 and its conjugates in vivo. **a** HEK293 cells expressing WT or Cys-free FLAG-tagged FAT10 were treated for 0.5, 1, or 2.5 h with 50 μg/mL CHX to monitor degradation rates. In addition, cells were treated with 10 μM proteasome inhibitor MG132 for 3.5 h, as indicated. Cells were harvested, lysed, and subjected to immunoprecipitation using EZview Red anti-FLAG-M2 affinity gel. Proteins were separated on 4–12% NUPAGE gradient gels and subjected to western blot analysis using a directly peroxidase-labeled, monoclonal FLAG-reactive antibody (clone M2). β-actin was used as loading control. One representative experiment out of four independent experiments with similar outcomes is shown. **b** ECL signals were quantified and graphs show the amount of monomeric as well as of conjugated WT or Cys-free FLAG-FAT10 normalized to the amount of β-actin in the lysate. The values of the untreated cells were set to 100% and the other values were calculated accordingly. Shown are the values of four independent experiments with similar outcomes as means ± s.e.m. **c** Molecular dynamics (MD) simulations of the N- (left) and C-domain (right) of human FAT10. Top panels: three of the most prominent structures obtained by root mean square deviation clustering from each MD simulation in cartoon representation (WT: white, Cys-free: blue). Bottom panels: Residue-wise root mean square fluctuation (RMSF) values. Arrows in the graph show position of cysteine mutations. N- and C-terminal tails are flexible. Moreover, flexibility is decreased in the N-domain in Cys-free mutants while the overall structure of both domains is preserved

than that of many other cellular proteins[30]. This raises the question as to why and how FAT10 is degraded so rapidly. Previous attempts to determine the high-resolution structure of FAT10 were hampered by the poor solubility of FAT10 in mammalian cells or when overexpressed in bacterial cells[21,22]. Here, we show that FAT10 has a melting temperature of only 41 °C that may explain its poor stability. We have succeeded in improving the solubility of FAT10 by replacing its Cys residues. This has not only allowed us to determine the 3D structures of the FAT10 N- and C-domains (Figs. 1 and 2), but also revealed that these mutations dramatically stabilized FAT10 in cellulo without affecting its conjugation to substrates (Fig. 5a, b). However, we cannot rule out that the inserted mutations apart from stabilizing FAT10 also affect FAT10-mediated degradation by other means. While Ub adopts a compact fold with the very N-terminus of Ub already being part of the Ub β-sheet, FAT10 contains a seven amino acid N-terminal extension that is intrinsically disordered and accelerates FAT10-mediated degradation (Fig. 6c, d). We also found that, in contrast to Ub (Fig. 6a), a FAT10-GFP model substrate and FAT10 itself were degraded by the proteasome even in the presence of VCP inhibitors (Fig. 6b–d). This VCP-independent degradation required the presence of the N-terminal disordered tail of FAT10, since its deletion rendered FAT10 degradation sensitive to VCP inhibition (Fig. 6c, d). These results strongly suggest that the unstructured N-terminal extension may help to initiate proteasomal degradation of FAT10, perhaps by making contact with the ATPase ring of the 26S proteasome[35,36] and by pulling FAT10 along with its substrate into the ATPase ring for further unfolding and proteolysis. Of note, the importance of the disordered N-terminal tail for fast and direct FAT10 degradation agrees well with the idea that Ub-mediated degradation by the 26S proteasome not only relies on a covalently attached poly-Ub tag, but also on loosely folded segments within a substrate protein which the ATPase ring of the proteasome can grasp. Such segments are generated by VCP upstream of the proteasome for tightly folded proteins[24].

In conclusion, the long awaited three-dimensional structure of FAT10 at high resolution reveals that in contrast to proteasomal targeting via the tightly folded and well-soluble ubiquitin, FAT10-dependent degradation relies on a flexible N-terminal heptapeptide and on lose folding of its UBDs. Given that FAT10 is not recycled but degraded along with its substrates, these two modifier-specific hallmarks make FAT10 an irreversible and efficient degradation signal which operates independently of VCP activity. These unique traits of degradation targeting may be advantageous in situations of inflammation and for mounting an immune response against invading pathogens.

## Methods

**Cloning and constructs for structural studies**. N-terminally truncated FAT10 (amino acids 5–165; WT ΔN), Cys-free variants of full-length (FL) human FAT10 (C7T, C9T, C134L, C162S), and N-terminally truncated FAT10 (amino acids 5–165; ΔN) were cloned into pETM-41 vectors (EMBL Heidelberg) to express His₆-MBP-fusion proteins, while the Cys-free FAT10 N-domain (amino acids 5–86; C7T, C9T) was cloned into a pETM-30 vector (EMBL Heidelberg) to express a His₆-GST-fusion protein, and the Cys-free FAT10 C-domain (amino acids 85–165; C134L, C160S, C162S) into a pETM-10_60 vector to express a His₆-NusA-His₆-fusion protein. Details on primers used in this study are provided in Supplementary Table S3. The RPN10 UIM2 was cloned into a pRTDuet vector to express a His₆-GB1-fusion protein. All bacterial expression constructs except for the RPN10 UIM2 construct contain a TEV protease cleavage site C-terminal to the purification tag that results after cleavage in an additional Gly residue at the N-terminus of the N-terminally truncated FL protein and the N-domain, an additional N-terminal GA sequence for the FL FAT10 and the N-domain, and an N-terminal GAMG sequence for the C-domain.

**Protein expression and purification**. All recombinant proteins were expressed in *Escherichia coli* BL21(DE3) CodonPlus cells (Stratagene) in LB (for crystallographic studies) or M9 minimal medium (for NMR studies) containing ¹³C-glucose and/or ¹⁵NH₄Cl as sole sources of carbon and nitrogen, respectively. All recombinant proteins used for structural and biophysical characterization were lysed by sonication in lysis buffer (50 mM sodium phosphate (pH 7.5), 300 mM NaCl, 10 mM imidazole, and 1 mM DTT). Cell debris was removed by centrifugation and the filtered supernatant applied to Ni-NTA beads (Quiagen) for affinity chromatography. Purified proteins were eluted with lysis buffer containing 300 mM imidazole. The elution fractions were supplemented with His-tagged TEV protease (except for the RPN10 UIM2) and dialyzed in dialysis buffer (50 mM sodium phosphate (pH 7.5), 300 mM NaCl, and 1 mM DTT). After cleavage, the desired proteins were separated from His₆-tagged cleavage products and His₆-tagged TEV protease by Ni-affinity chromatography using dialysis buffer. All proteins were further purified by size-exclusion chromatography. Pure proteins for NMR and DSF analyses were buffer exchanged to NMR buffer (20 mM sodium phosphate pH 7.5, 150 mM NaCl) that contained 1 mM DTT for proteins containing Cys residues. For crystallization, proteins were in 20 mM HEPES (pH 7.5), 150 mM NaCl.

**Crystallization and structure solution with refinement**. The Cys-free FAT10 N-domain was crystallized at a protein concentration of 10 mg/ml in 3.2 M (NH₄)₂SO₄, 0.1 M citric acid (pH 4.0). Diffraction data were collected at 100 K using a wavelength of 1 Å and a PILATUS 6M-F detector at the beamline PXII of the Swiss Light Source (PSI, Villigen, Switzerland). Data were processed using XDS[37] and molecular replacement was performed using Phaser[38]. The structure was finalized by iterative manual modeling with Coot[39] and refinement with Phenix[40]. In each asymmetric unit, three protein chains could be visualized. In total, 98.6% of all residues were in favored regions and 1.4% of all residues were in additional allowed regions of the Ramachandran plot. No outliers were found. Ramachandran plot statistics were obtained using the MolProbity server (http://molprobity.biochem.duke.edu/). All figures displaying protein structures were generated with PyMOL (http://www.pymol.org/).

**NMR methods and structure calculation**. All NMR experiments were conducted on FAT10 samples at 25 °C on 500, 600, or 800 MHz Bruker Avance spectrometers equipped with room temperature probe heads. ¹H, ¹³C, and ¹⁵N chemical shifts were assigned by standard methods[41] at 1 mM, 1.2 mM, and 2 mM concentration of the N-domain, the C-domain, and ΔN FAT10, respectively. Steady-state {¹H}-¹⁵N NOE values were measured at 1.1 mM concentration in an interleaved fashion by standard methods[42] and calculated as peak intensity ratio with or without saturation. Means and standard deviations were estimated from three independent experiments. Distance restraints for the FAT10 C-domain were derived from 3D

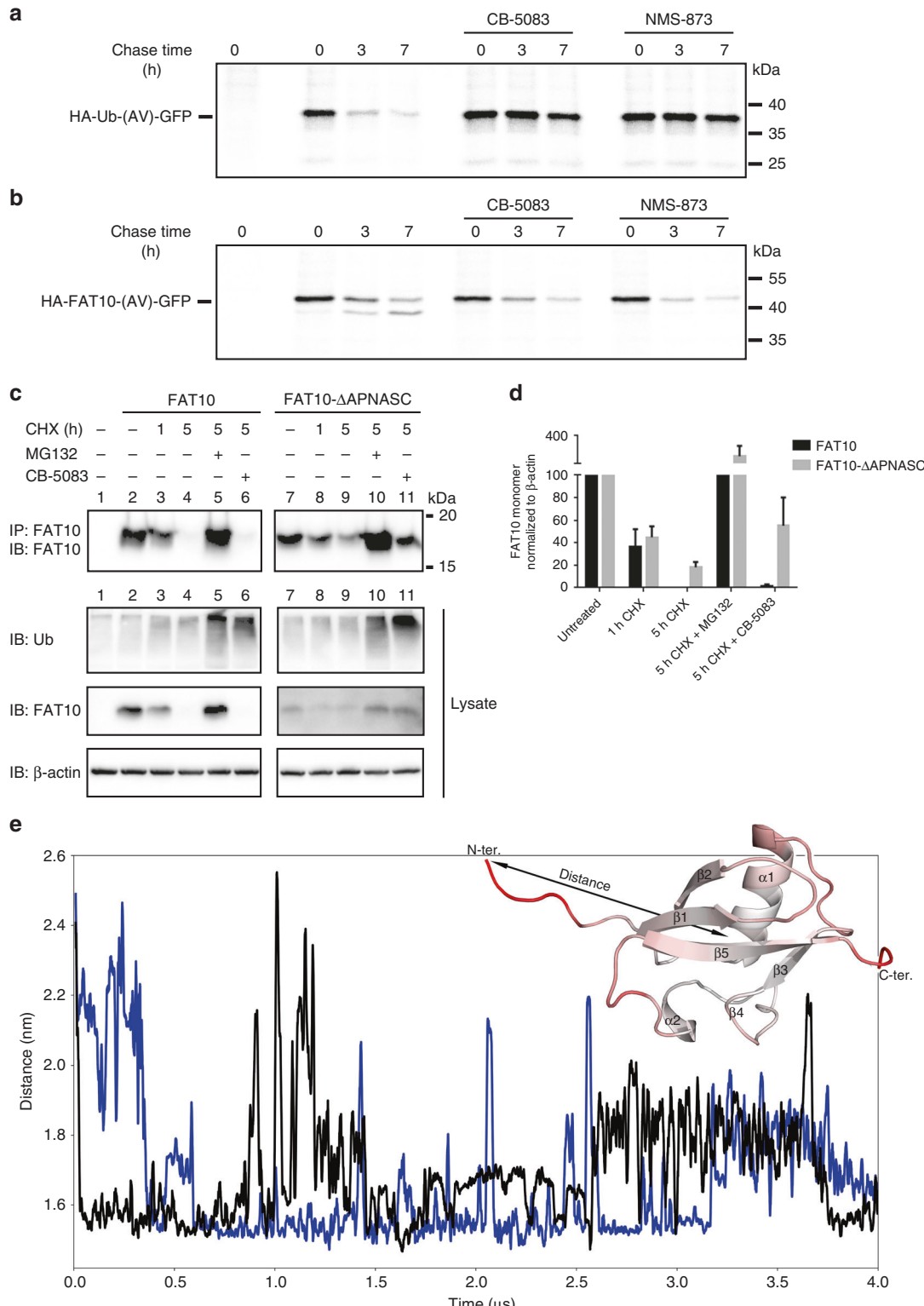

HNH-, HCH- and CNH- and 2D NOESY spectra[43] using a 1.2 mM sample recorded at 800 MHz using a mixing time of 120 ms. Dihedral angle restraints for backbone $\varphi$ and $\psi$ angles were derived from chemical shifts using TALOS[44]. Hydrogen bonds were identified in NOESY spectra and set to 1.8 Å for $H^N$–O and 2.8 Å for N–O distances. All spectra were processed with the NMRPipe/NMRDraw package[45] and analyzed with CARA/XEasy[46]. Structures were calculated using ARIA1.2[47]. Ramachandran plot statistics were obtained with PROCHECK-NMR using residues 87–160 of the ten lowest energy struictures. 90.2% of all residues were in favored regions and 9.8% of all residues were in additional allowed regions

of the Ramachandran plot. No outliers were found. Chemical shift perturbation studies were performed using $^1H,^{15}N$-HSQC spectra recorded at a protein concentration of 225 μM for the N-domain, 1.2 mM for the C-domain, and 700 μM for ΔN FAT10.

**Sequence alignment.** Multiple sequence alignments were performed with MUSCLE[48] and displayed with Jalview[49]. Sequence conservation was calculated using the ConSurf server[50], while residue-specific solvent accessibilities (SASA) were

**Fig. 6** The N-terminus of FAT10 is crucial for its VCP-independent degradation. **a, b** HeLa cells were transiently transfected with HA-ubiquitin-(AV)-GFP or HA-FAT10-(AV)-GFP expression constructs. After 16 h cells were metabolically labeled with $^{35}$S-methionine for 1 h and chased in the presence of 5 μM of VCP inhibitors CB-5083 or NMS-873 for the indicated time periods. Then, cells were lysed and HA-tagged proteins were immunoprecipitated, separated by SDS-PAGE, and visualized by autoradiography. **c** HEK293 cells were transiently transfected with expression constructs for WT untagged FAT10 or a FAT10 variant lacking the six N-terminal amino acids APNASC (FAT10-ΔAPNASC). 24 h after transfection, cells were treated with 50 μg/mL CHX in the presence or absence of 10 μM proteasome inhibitor MG132 or 10 μM VCP inhibitor CB-5083 for 5 h. Cleared lysates were subjected to immunoprecipitation for FAT10 and degradation of FAT10 was visualized by western blot. Endogenous ubiquitin in total lysates was monitored by western analysis as well (IB: Ub). β-actin served as loading control. One representative experiment out of three experiments with similar outcomes is shown. **d** ECL signals of immunoprecipitated FAT10 and FAT10-ΔAPNASC were quantified and normalized to signals of the loading control β-actin. The values of untreated cells were set to 100% and the other values were calculated accordingly. The mean of three independent experiments with similar outcomes is shown as means ± s.e.m. **e** Molecular dynamics simulations of the N-terminal ubiquitin-like domain (UBD) of human FAT10 (WT) showing that the N-terminal unstructured region of FAT10 is highly flexible. The graph shows the distance between the N-terminus and center of the N-domain during the course of two simulations with different protonation states of histidine (deprotonated: black, protonated: blue). The flexible N-terminus forms contacts with the domain but is repeatedly released in both simulations. Inset: N-domain in cartoon representation colored in red according to high RMSF values. The arrow indicates the distance shown in time series

obtained with ASA-View[51]. The Ub binding score was calculated by analyzing the residue-specific buried SASAs of 16 different Ub–protein complexes (PDB-IDs: 1Q0W, 4UN2, 1OTR, 1S1Q, 2KWV, 3LDZ, 2MUR, 1WRD, 3CMM, 4MDK, 4LJ0, 3OLM, 5CAW, 2XBB, 3TMP, 5CRA) with the PISA server[52]. All incidents of buried surface SASAs > 10% were scored and normalized to a scale of 0 to 1 for each residue in Ub.

**Primers and constructs for cell culture.** Plasmids used for transient transfection of HEK293 cells (originally purchased from ATCC) were pcDNA3.1-His-3xFLAG-FAT10[9], pcDNA3.1-HA-USE1[11], and pcDNA3-FLAG-ubiquitin (kindly contributed by M. Basler, University of Konstanz, Germany). Constructs for the expression of FAT10 linker mutants were generated by site-directed mutagenesis using pcDNA3.1-His-3xFLAG-FAT10 as a template and the following primer pairs (for sequences see Supplementary Table 3): AA-363 and AA-364 for the generation of pcDNA3.1-His-3xFLAG-FAT10-5xAla; AA-361 and AA-362 for the generation of pcDNA3.1-His-3xFLAG-FAT10-5xPro; AA-367 and AA-368 for the generation of pcDNA3.1-His-3xFLAG-FAT10-5xGly; AA-369 and AA-370 for the generation of pcDNA3.1-His-3xFLAG-FAT10-Δlinker. Wildtype *Fat10* cDNA without a tag, was cloned into pcDNA6.1-myc-His (Invitrogen) to generate pcDNA6.1-FAT10 by restriction digest of pcDNA3.1-His-3xFLAG-FAT10 using restriction enzymes EcoRI and NotI and subsequent ligation of the coding sequence of *Fat10* into EcoRI/NotI digested pcDNA6.1-myc-His, thereby introducing a stop codon 5′ of the myc-His-tag. For the expression of the tagless, N-terminally shortened FAT10 version FAT10ΔAPNASC, the FAT10ΔAPNASC coding sequence was generated by PCR with pcDNA3.1-His-3xFLAG-FAT10 as template and primers 5′-EcoRI-FAT10 dAPNASC fwd and 3′-NotI-FAT10 dAPNASC rev and ligated into EcoRI/NotI digested pcDNA6.1-myc-His, with a stop codon 5′ of the myc-His-tag. Expression constructs for FAT10-ubiquitin hybrid mutants (pcDNA3.1-His-3xFLAG-FAT10-Ub and -FLAG-Ub-FAT10) were generated by overlap extension PCR with pcDNA3.1-His-3xFLAG-FAT10 as template. Exchange of the N-terminal UBL domain of FAT10 in pcDNA3.1-His-3xFLAG-FAT10 to generate pcDNA3.1-His-3xFLAG-Ub-FAT10 was performed as follows: pRK5-HA-Ubiquitin(K0) (Addgene) was used as a template for the amplification of the coding sequence of Ub(K0) with primers AA-371 and AA-372, whereas the codons for the diglycine motif of Ub(K0) were not included in the primer. AA-371 and AA-372 contained sequences which were overlapping with the His-3xFLAG-tag and the FAT10 linker as well as parts of the FAT10 C-terminal UBL domain, respectively. Exchange of the C-terminal UBL domain of FAT10 in pcDNA3.1-His-3xFLAG-FAT10 to generate pcDNA3.1-His-3xFLAG-FAT10-Ub was also performed with pRK5-HA-Ubiquitin(K0) as template and primers AA-373 and AA-374. These primers contained overlapping sequences of the FAT10 linker region and parts of the N-terminal UBL domain of FAT10 in case of AA-373, and overlapping sequences of the 3′ non-translated region of pcDNA3.1-His-3xFLAG-FAT10 in case of AA-374. The amplicons of the two independent PCR reactions were then used as primers for a subsequent site-directed mutagenesis with pcDNA3.1-His-3xFLAG-FAT10 as template to generate pcDNA3.1-His-3xFLAG-Ub(K0)ΔGG-FAT10 and pcDNA3.1-His-3xFLAG-FAT10-Ub(K0), respectively. The sequences of all plasmids were verified by sequencing (Microsynth AG, Balgach, Switzerland). The resulting proteins are named in the text as FLAG-FAT10 in case of pcDNA3.1-His-3xFLAG-FAT10, FLAG-Ub-FAT10 in case of pcDNA3.1-His-3xFLAG-Ub(K0)ΔGG-FAT10 and FLAG-FAT10-Ub in case of pcDNA3.1-His-3xFLAG-FAT10-Ub(K0).

pcDNA3.1-His-3xFLAG-FAT10-(C0)-C134L was generated by PCR using pSUMO-6His-HA-FAT10-(C0)-C134L as template and primers AA-345 and AA-346. The PCR product was inserted via restriction sites EcoRI and NotI into pcDNA3.1-His-3xFLAG-FAT10[9], where wildtype FAT10 had been removed by restriction digest with the same enzymes. pSUMO-HA-FAT10-(C0)-C134L was generated as described in Supplementary Table 3.

**Cell culture and immunoprecipitation.** The human embryonic kidney cell line HEK293 was purchased from ATCC and used because FAT10 is strongly induced in them with IFN-γ and TNF leading to FAT10 conjugate formation[11]. HEK293 cells have been authenticated morphologically and by MHC typing and were shown to be free of mycoplasma using the MycoAlert™ kit (Roche). HEK293 cells were cultivated in Iscove's Modified Dulbecco's Medium (IMDM) supplemented with 10% fetal calf serum, 1% ultraglytamine, and 1% penicillin/streptomycin. Cells were transfected using TransIT-LT1 transfection reagent (Mirus) as described by the manufacturer's instructions. Cycloheximide chase experiments were performed as described in ref. [18]. Proteasome inhibitor MG132 (Enzo Lifesciences) or VCP inhibitor CB-5083 (Selleckchem) were added 6 h prior to harvesting with final concentrations of 10 μM each or in parallel to cycloheximide (CHX) (Sigma), which was added at the time points indicated with a final concentration of 50 μg mL$^{-1}$. Cells were harvested and lysed as described earlier[18]. Cleared lysates were subjected to anti-FLAG immunoprecipitation using EZview Red Anti-FLAG-M2 Affinity Gel (Sigma) for at least two hours at 4 °C or incubated over night with a FAT10-reactive monoclonal antibody (clone 4F1[11]) bound to protein A sepharose. Samples were washed twice with NET-TN (50 mM Tris-HCl pH 8.0, 650 mM NaCl, 5 mM EDTA, 0.5% Triton X-100) and twice with NET-T wash buffer (50 mM Tris-HCl pH 8.0, 150 mM NaCl, 5 mM EDTA, 0.5% Triton X-100). Proteins were separated on 12.5% Laemmli SDS gels and subjected to western blot analysis using monoclonal anti-FLAG M2-peroxidase-conjugated antibody (Sigma, cat. no. A8592-5X1MG 1:3000), polyclonal anti-USE1 antibody (1:1000)[11], polyclonal anti-poly-ubiquitin antibody (Dako, 1:2000), a polyclonal FAT10 reactive antibody (1:1000)[14], or a monoclonal antibody reactive against β-Actin (Abcam, cat. no. ab6276, 1:5000) as loading control.

Denaturing lysis was performed as previously described[53] with the following changes. Confluent cells of a 15 cm cell culture dish were washed once in PBS and directly lysed in 250 μl 2× lysis buffer (1× PBS, 2% SDS, 10 mM EDTA pH 8.0, 10 mM EGTA pH 8.0, and 1× protease inhibitor (cOmplete™ mini EDTA-free protease inhibitor cocktail, Roche)). Lysates were sonicated twice and supplemented with 50 μl 1 M DTT and boiled. Renaturation was performed by diluting the boiled samples in ten volumes of RIPA lysis buffer (150 mM NaCl, 50 mM Tris-HCl, pH 8.0, 1% Triton X-100, 0.5% Na-Deoxycholat, 0.1% SDS and 1× protease inhibitor (cOmplete™ mini EDTA-free protease inhibitor cocktail, Roche)) on ice. Lysates were cleared by centrifugation and subjected to immunoprecipitation and western blot analysis as described above.

**Pulse-chase experiments.** Transfections of the HA-ubiquitin-(AV)-GFP and HA-FAT10-(AV)-GFP constructs into HeLa cells (originally purchased from ATCC), metabolic labeling, pulse chase experiments, lysis, and immunoprecipitation was performed as follows. In total, $5 \times 10^6$ HeLa cells were transfected with 6 μg of the plasmids HA-Ubiquitin-(AV)-GFP or HA-FAT10-(AV)-GFP using 18 μg of polyethylenimine. 16 h after transfection the cells were washed with PBS and starved for 1 h in Met/Cys-free RPMI 1640 medium (Sigma) supplemented with L-glutamine, penicillin/streptomycin, and 10% dialyzed fetal calf serum, followed by labeling for 1 h with 0.25 mCi/ml [$^{35}$S]Met/Cys (Translabel, Hartmann Analytic). Subsequently, cells were washed three times with PBS, aliquoted, and chased for 0, 3, and 7 h in DMEM-Medium supplemented with 10% fetal calf serum in the absence or presence of VCP inhibitors ( 5 μM CB-5083 or  5 μM NMS-873). The labeled cells were harvested and lysed in 20 mM Tris/HCl, pH 8.0, 0.1% Triton X-100 for 30 min on ice. After centrifugation for 15 min at 15,000 × g, 1 volume of 20 mM Tris/HCl, pH 8.0, 300 mM NaCl was added to the supernatant, and an aliquot was analyzed with a β-counter. Equal amounts of radioactivity were used for immunoprecipitation with 25 μl of EZview red anti-HA affinity Gel (Sigma). The samples were incubated overnight at 4 °C with agitation. After five washes with the lysis buffer containing 300 mM NaCl, the immunoprecipitate was boiled in SDS sample buffer and analyzed by running a 12% SDS-PAGE, followed by autoradiography on a BioRad PMI personal molecular imager.

**CRISPR/Cas9 knockdown of USE1 in HEK293 cells.** HEK293 cells were transfected with pCMV-Cas9-GFP containing USE1-specific gRNA (Sigma). 24 h after transfection, single GFPhigh cells were sorted using BD FACS AriaTM Ilu (BD Biosciences) and cultivated in Iscove's Modified Dulbecco's Medium (IMDM) supplemented with 10% fetal calf serum and 1% penicillin/streptomycin. The successful knockout of USE1 was verified by western blot analysis using a USE1-reactive polyclonal antibody[11].

**Quantification of enhanced chemiluminescence signals.** Enhanced chemiluminescence (ECL) signals were quantified with Image Lab 4.1 software (BioRad). Signals of monomeric FAT10 in the lysate or of bulk FAT10 conjugates upon immunoprecipitation from at least three independent experiments were calculated (excluding monomeric FAT10 in the immunoprecipitation) and normalized to the ECL signals of the loading control β-actin. Values in the figures are given as means ± s.e.m. Values of untreated FLAG-FAT10, FAT10 or FAT10ΔAPNASC monomers or conjugates were set to unity and the other values were calculated accordingly.

**Differential scanning fluorimetry.** Thermal protein unfolding was monitored as intrinsic fluorescence changes with temperature measured at wavelengths of 330 and 350 nm on a Prometheus NT.48 instrument (NanoTemper). The nanoDSF Standard Grade capillaries contained 10 μL of 100 μM WT ΔN FAT10, C0 ΔN FAT10 or ubiquitin in 20 mM NaP pH 7.5, 150 mM NaCl, 1 mM DTT. Three independent measurements were performed per sample. The temperature was varied in 0.036 °C steps from 20 to 95 °C. Melting temperatures ($T_m$ values) were determined from the first derivative curves using the PR.Control software (NanoTemper).

**Molecular dynamics simulations.** All molecular dynamics (MD) simulations were performed with the GROMACS simulation package v5.1.4[54] with the GROMOS96 54a7 force field[55] and the SPC/E water model. Temperature and pressure were kept at 300 K and 1 bar using the velocity rescaling thermostat and the Parrinello–Rahman barostat, respectively. The Verlet cut-off scheme was applied as implemented in GROMACS. The LINKS algorithm was used to constrain all bonds. The default md (leap-frog) integrator was used with an integration time step of 2 fs and a cut-off for short range van der Waals interactions of 1.4 nm. Electrostatic interactions were treated with the Particle Mesh Ewald scheme and a 1.4 nm cut-off. Initial conformations for simulations of FAT10 N- and C-domains were generated from the experimental structures determined in this study. Reference simulations of Ub was started from PDB-ID: 1UBQ. All structures were relaxed by energy minimization and equilibrated after solvation. Equilibration was achieved in three short runs of 200 ps: (1) under constant temperature (NVT) with a position restrained backbone; (2) under constant temperature and pressure (NPT) with a position restrained backbone; (3) NPT without any position restrains. Production runs were performed for 4 μs. Simulations were performed with protonation states according to pH 5 (for the N-domain a separate simulation was performed with deprotonated histidine sidechains (pH 7) to exclude a potential structural impact of pH 5). The following MD simulations were performed for this study: FAT10 N-domain with WT, WT (His neutral), and Cys-free sequence; FAT10 C-domain with WT and Cys-free sequence, and Ub with WT sequence.

**Data availability.** Coordinates, structure factors and chemical shift assignments have been deposited in the Protein Data Bank and the Biological Magnetic Resonance Data Bank under PDB accession codes 6GF1 and 6GF2 and BMRB accession codes 27467 and 27527. Other data supporting the findings of this manuscript are available from the corresponding authors upon reasonable request.

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

## Acknowledgements

We thank Magnus Jäckl for collecting the crystallographic data of the FAT10 N-domain and Michael Braun for help with crystal structure refinement. Sebastian Fischer is acknowledged for help with mutagenesis and protein production. We are grateful to Milos Cvetkovic for performing chemical shift perturbation experiments with Ub and the RPN10 UIM2 and to Gernot Längst for permitting us to use the Prometheus NT.48 instrument. Michael Basler is acknowledged for a plasmid contribution. This study was supported by the German Research Foundation (DFG) Collaborative Research Center SFB969, projects C01 (M.G.) and B09 (C.P.), the Velux Foundation (grant 1029) (A.A. and M.G.) and the Max Planck Society (S.W.). J.B. received a stipend from the Graduate School Chemical Biology at the University of Konstanz. Simulations were performed on computational resources funded within the bwHPC program by the state of Baden-Württemberg and the DFG (INST 35/1134-1 FUGG).

## Author contrubutions

A.A., S.A., N.C., P.R., S. Stotz, A.B., R.S., S. Scheuermann, J.B., M.C.S.-S., G.S., and S.W. performed experiments, evaluated, and refined data. A.A., M.G., and S.W. wrote the manuscript. C.P., M.G., and S.W. corrected and refined the manuscript. A.A., C.P., M.G., and S.W. conceived and supervised experiments and acquired funding.
