## [Peer Review File · Nature Communications]

Reviewers' comments:

Reviewer #1 (Remarks to the Author):

Groettrup and co-workers present the long illusive structure of full-length FAT10. However the most important contribution is that they authors has established the role for N-terminal unfolded region of FAT10 in the unfolding of substrates for degradation by the proteasome.

Reviewer #2 (Remarks to the Author):

This paper reports on the structure of the FAT10 protein which has two UBL domains. The protein was stabilized by making a number of mutations to the cys residues in the sequence. This manuscript suggests that the FAT10 UBLs are more stable than Ub, and that this results in its efficient degradation in vivo. Furthermore, the flexible N-terminus drives fast and direct FAT10 degradation.

I disagree with the author's suggestion that a lower intrinsic stability for the FAT10 domains compared to Ub, explains its in vivo stability. Though the MD simulations indicate that the FAT10 domains are more flexible than Ub, this is not backed up experimentally. Also, clearly linking intrinsic stability to in vivo stability is not trivial.

Intro: avoid a.k.a. throughout manuscript.

References for statements regarding UBA6 and USE1 should be added.

On page 3, avoid a.k.a. Its yeast S5a, human Rpn10.

References missing for NEDD8 ultimate buster.

Page 5: "we detected only *few* peaks" the authors mean small magnitude or negative NOEs.

Page 6: "To characterize the structures of the FAT10 domains in more detail, we crystallized the Ndomain and solved the solution NMR structure of the C-domain" I think the authors should indicate if the C-domain did not crystallize, though this is implied in the next sentence.

Page 6: too many digits in the error and value for the rmsd value.

Page 6: aa should be clearly defined as amino acid throughout the manuscript.

Page 7: "dense aliphatic network" should probably be hydrophobic network.

Page 7: "**even* the lengths of the loops are conserved" implies a comparison which is not clear as written.

Page 7: "...structures of the FAT10 N- and C-domain"s" in hand.

Page 8: ...conserved surfaces on the FAT10 C-domain included *those* next to the essential C-terminal...

For Supplementary Figures 5 and 6, the authors indicate that experiments are done in triplicate, but do not show data for all experiments.

Page 9: ...were expressed at different level"s"...

Page 9: Figure 3, ECL is not defined, I find a number acronyms are not defined.

Page 9: For Figure 3c, the data only roughly support the suggestion that conjugates are degraded at similar rates, the errors are quite large. The initial drop at 2.5 hrs suggests that the degradation is similar. The authors should clarify this in the manuscript.

Page 11, top: the Ub/FAT10 chimera experiments are interpreted as though the stability of Ub results in a slower rate of degradation in the CHX/MG132 experiments. The authors point to atomic fluctuations and lower secondary structure content for the FAT10 domains compared to Ub, observed in typically 4 microsecond MD simulations. They also perform MD simulations of a Cys-free/C134L mutant, which suggest it is more rigid than wildtype FAT10. These theoretical studies are not backed up by biophysical measurements of protein dynamics or protein stability. Thus, they raise intriguing possibilities, but fall short of demonstrating clear differences in stabilities for the FAT10 domains compared to Ub and FAT10 mutants. To that end, I do not agree with the authors' assertions that the biochemical experiments reported on page 11 combined with MD simulations "support the idea that the native FAT10 protein may be rather unstable and thus prone to fast and direct proteasomal degradation".

Page 12: GFP-Ub substrates have been developed previously by the Deshaies group, this should be referenced.

Page 14: It is not clear what "this list" refers to in the first sentence of the second paragraph.

Page 16: I do not find the conclusions to be fully supported by the data, as described in comments regarding page 11 above.

Point-to-point reply to reviewers' and editorial recommendations and points of criticism for the manuscript by Aichem et al. entitled: 'The structure of the ubiquitin-like modifier FAT10 reveals a novel targeting mechanism for proteasomal degradation' NCOMMS-18-11630-T

Please also address the following editorial requests:

** Please state that you determined the structures of the two FAT10 domains individually in the Abstract.*

Reply 1: We thank the editor for this comment and have clarified this in the abstract of the revised manuscript. We should also mention that we have shortened the previous title 'The structure of the ubiquitin-like modifier FAT10 reveals a novel targeting mechanism for degradation by the 26S proteasome' to 'The structure of the ubiquitin-like modifier FAT10 reveals a novel targeting mechanism for proteasomal degradation' in order to meet the 15 word limit for titles.

** The r.m.s.d. for bond lengths is high with 0.023 Å: please explain why.*

Reply 2: We very much appreciate the Editor's comment. We used a low value for the geometry weighting term during structure refinement explaining the relatively high r.m.s.d. for bond lengths. We have now refined the structure with tighter geometry restraints. The r.m.s.d. for bond lengths now equals 0.007 Å. We have updated the structure statistics table (Supplementary Table 1) with the values of the new, improved refinement.

Reviewers' comments:

Reviewer #1 (Remarks to the Author):

Groettrup and co-workers present the long elusive structure of full-length FAT10. However the most important contribution is that they authors has established the role for N-terminal unfolded region of FAT10 in the unfolding of substrates for degradation by the proteasome.

Reply 3: We thank Reviewer #1 for his/her positive comments and for acknowledging the novelty of our findings.

Reviewer #2 (Remarks to the Author):

This paper reports on the structure of the FAT10 protein which has two UBL domains. The protein was stabilized by making a number of mutations to the cys residues in the sequence. This manuscript suggests that the FAT10 UBLs are more stable than Ub, and that this results in its efficient degradation in vivo. Furthermore, the flexible N-terminus drives fast and direct FAT10 degradation.

I disagree with the author's suggestion that a lower intrinsic stability for the FAT10 domains compared to Ub, explains its in vivo stability. Though the MD simulations indicate that the FAT10 domains are more flexible than Ub, this is not backed up

experimentally. Also, clearly linking intrinsic stability to in vivo stability is not trivial.

Reply 4: We thank Reviewer #2 for the critical reading of our manuscript and the numerous useful suggestions. To provide experimental evidence as to the instability of FAT10 in comparison to ubiquitin (Ub) we performed differential scanning fluorimetry experiments and show that WT FAT10 and the Cys-free FAT10 mutant have melting temperatures of only 41 °C and 47 °C, respectively, while ubiquitin melts at 83 °C. These results thus show that FAT10 is significantly less stable than Ub. We have added these results in Figure 4a and Supplementary Fig. 7a. These data are in agreement with the molecular dynamics simulation measurements shown in Fig. 5c, which show that the Cys-free FAT10 mutant is less flexible than the WT variant. Since the degradation rates of monomeric FAT10 and FAT10 conjugates are slower for the Cys-free as compared to WT FAT10 (Fig. 5a,b) we think we have evidence that the lower stability of FAT10 contributes to the high speed of FAT10 degradation. However, we concede in the discussion that the point mutations introduced into the FAT10 sequence for obtaining the Cys-free variant may in addition to stabilizing FAT10 affect FAT10-mediated degradation by other means.

Intro: avoid a.k.a. throughout manuscript.

Reply 5: We apologize for using this apparently not very common abbreviation and have spelled it out throughout the manuscript.

References for statements regarding UBA6 and USE1 should be added.

Reply 6: We have added the respective references Chiu et al. (2007) Mol. Cell 27:1014-1023; Jin et al. (2007) Nature 447:1135-1137; Pelzer et al. (2007) J. Biol. Chem. 282:23010-23014; Aichele et al. (2010) Nat. Commun. 1:13 to the revised manuscript.

On page 3, avoid a.k.a. Its yeast S5a, human Rpn10.

Reply 7: We apologize for being unclear and have clarified this in the revised manuscript on p. 3.

References missing for NEDD8 ultimate buster.

Reply 8: We have inserted here the references Hipp et al. (2004) J. Biol. Chem. 279:16503-16510 and Schmidtke et al. (2006) J. Biol. Chem. 281:20045-20054.

*Page 5: “we detected only *few* peaks” the authors mean small magnitude or negative NOEs.*

Reply 9: We apologize for the imprecise description and have corrected this in the revised manuscript on p. 5.

Page 6: “To characterize the structures of the FAT10 domains in more detail, we crystallized the Ndomain and solved the solution NMR structure of the C-domain” I think the authors should indicate if the C-domain did not crystallize, though this is implied in the next sentence.

Reply 10: We appreciate the Reviewer's comment and have made clear on p. 6 that the C-domain did not crystallize.

Page 6: too many digits in the error and value for the rmsd value.

Reply 11: We have reduced the number of digits in the rmsd values on p. 6.

Page 6: aa should be clearly defined as amino acid throughout the manuscript.

Reply 12: We have replaced "aa" with "amino acid" throughout the text of the revised manuscript.

Page 7: "dense aliphatic network" should probably be hydrophobic network.

Reply 13: We apologize for being unclear and have revised the corresponding text on p. 7 as follows: "stabilized by a dense hydrophobic interaction network that is almost void of aromatic residues".

*Page 7: "*even* the lengths of the loops are conserved" implies a comparison which is not clear as written.*

Reply 14: We clarified this sentence on p. 7 of the revised manuscript as follows: "Overall, comparison of the FAT10 N-domain structure with Ub showed that not only the structured regions are well-conserved, but also the lengths of the loops except for the α 2- β 5 loop that contains two additional residues in the N-domain (**Fig. 1f, 2a**)."

Page 7: ...structures of the FAT10 N- and C-domain"s" in hand.

Reply 15: We apologize for the oversight and corrected this on p. 7 of the revised manuscript.

*Page 8: ...conserved surfaces on the FAT10 C-domain included *those* next to the essential C-terminal...*

Reply 16: We have inserted the word 'those' in this sentence as suggested.

For Supplementary Figures 5 and 6, the authors indicate that experiments are done in triplicate, but do not show data for all experiments.

Reply 17: We have quantitatively evaluated the enhanced chemiluminescence signals for all three western blots of Supplementary Fig. 5 and the three western blots for Supplementary Fig. 6 and have graphically depicted the results with mean values and S.E.M. in Supplementary Figure 5b and Supplementary Figure 6b.

Page 9: ...were expressed at different level"s"...

Reply 18: This typo was corrected.

Page 9: Figure 3, ECL is not defined, I find a number acronyms are not defined.

Reply 19: We now have defined the abbreviation ECL which stands for ‘enhanced chemoluminescence’

Page 9: For Figure 3c, the data only roughly support the suggestion that conjugates are degraded at similar rates, the errors are quite large. The initial drop at 2.5 hrs suggests that the degradation is similar. The authors should clarify this in the manuscript.

Reply 20: We agree with our referee that wild type FAT10 (containing KPSDE linker) and all the analysed linker mutants of FAT10 and their conjugates are degraded at similar rates within 2.5 hours. We had performed similar CHX chase experiments with shorter chase times which confirmed this conclusion (data not shown). As requested, we have changed the description of the results in Fig. 3c,d to ‘All monomeric FAT10 variants (Fig. 3d, left panel) and their conjugates (Fig. 3d, right panel) were similarly degraded within 2.5 hours of CHX chase and their degradation could be blocked by proteasome inhibition.’ in order to clarify this conclusion.

Page 11, top: the Ub/FAT10 chimera experiments are interpreted as though the stability of Ub results in a slower rate of degradation in the CHX/MG132 experiments. The authors point to atomic fluctuations and lower secondary structure content for the FAT10 domains compared to Ub, observed in typically 4 microsecond MD simulations. They also perform MD simulations of a Cys-free/C134L mutant, which suggest it is more rigid than wildtype FAT10. These theoretical studies are not backed up by biophysical measurements of protein dynamics or protein stability. Thus, they raise intriguing possibilities, but fall short of demonstrating clear differences in stabilities for the FAT10 domains compared to Ub and FAT10 mutants. To that end, I do not agree with the authors assertions that the biochemical experiments reported on page 11 combined with MD simulations “support the idea that the native FAT10 protein may be rather unstable and thus prone to fast and direct proteasomal degradation”.

Reply 21: We appreciate the Reviewer’s comment. To provide biophysical measurements of the instability of FAT10 in comparison to Ub we performed differential scanning fluorimetry experiments and show that WT FAT10 and the Cys-free FAT10 mutant have melting temperatures of only 41 °C and 47 °C, respectively, while ubiquitin melts at 83 °C. These results thus show that FAT10 is significantly less stable than Ub. We show these new results in Fig. 4a and Supplementary Fig. 7a.

Page 12: GFP-Ub substrates have been developed previously by the Deshaies group, this should be referenced.

Reply 22: We are citing in the revised version the recent publication by the Deshaies group (Blythe et al. (2017) *PNAS* 114:E4380-4388) who have shown that the

degradation of Ub-(G76V)-GFP depends on VCP activity. This remark is well taken, because this publication was an important inspiration for our experiments.

Page 14: It is not clear what “this list” refers to in the first sentence of the second paragraph.

Reply 23: We have rephrased this sentence to make clearer and eliminated the allusion to the list of different properties of ubiquitin and FAT10 compiled in the paragraph before.

Page 16: I do not find the conclusions to be fully supported by the data, as described in comments regarding page 11 above.

Reply 24: See comments in replies 4 and 21.